# Estimating the effect of rainfall on the surface temperature of a tropical lake

Gabriel Gerard Rooney[1], Nicole van Lipzig[2], and Wim Thiery[3,4]

[1]Met Office, FitzRoy Road, Exeter, EX1 3PB, UK
[2]KU Leuven, Department of Earth and Environmental Sciences, Celestijnenlaan 200E, 3001 Leuven, Belgium.
[3]ETH Zurich, Institute for Atmospheric and Climate Science, Universitaetstrasse 16, 8092 Zurich, Switzerland
[4]Vrije Universiteit Brussel, Department of Hydrology and Hydraulic Engineering, Pleinlaan 2, 1050 Brussels, Belgium

**Correspondence:** G. G. Rooney (gabriel.rooney@metoffice.gov.uk)

**Abstract.** We make use of a unique high-quality, long-term observational dataset on a tropical lake to assess the effect of rainfall on lake surface temperature. The lake in question is Lake Kivu, one of the African Great Lakes, and was selected for its remarkably uniform climate and availability of multi-year, over-lake meteorological observations. Rain may have a cooling effect on the lake surface by lowering the near-surface air temperature, by the direct rain heat flux into the lake, by mixing the lake surface layer through the flux of kinetic energy, and by convective mixing of the lake surface layer. The potential importance of the rainfall effect is discussed in terms of both heat flux and kinetic-energy flux. To estimate the rainfall effect on the mean diurnal cycle of lake surface temperature, the data are binned into categories of daily rainfall amount. They are further filtered based on comparable values of daily mean net radiation, which reduces the influence of radiative-flux differences. Our results indicate that days with heavy rainfall may experience a reduction in lake surface temperature of approximately 0.3 K by the end of the day compared to days with light-to-moderate rainfall. Overall this study highlights a new potential control on lake surface temperature, and suggests that further efforts are needed to quantify this effect in other regions and to include this process in land-surface models used for atmospheric prediction.

## 1 Introduction

Lakes are important features of the terrestrial environment for physical, ecological, economic and recreational reasons. Physically, lake-atmosphere interactions can influence the local weather and climate. Thus their representation in earth-system modelling has increased in complexity in recent years. Lake water surface temperature (LWST) is of particular relevance to

atmospheric modelling due to the contrast in temperature, and hence in boundary-layer fluxes, that often exists between lakes and their surroundings (Mironov et al., 2010).

At high latitudes, correct prediction of freezing temperatures, and thus ice-cover periods, is important to obtain accurate boundary-layer fluxes. In the tropics, varying temperature contrasts between lakes and the surrounding land may be associated with cycles of severe weather. For instance, remote sensing data highlight an important impact of the African Great Lakes on the diurnal precipitation and thunderstorm cycle, especially over Lake Victoria and Lake Tanganyika (Camberlin et al., 2017; Thiery et al., 2017). During the afternoon, the typical tropical convective precipitation falls over land, but hardly any rainfall is observed over the lakes. At night, in contrast, very little precipitation is produced over land, while strong convection develops over the lakes, leading to high precipitation amounts. This phenomenon is caused by the diurnal cycle of the lake-land temperature difference, which leads to land breezes converging over the lake surface during the night. When these air masses, moistened by the lake, lift up into the atmosphere, they generate convective precipitation and often thunderstorms (Docquier et al., 2016; Thiery et al., 2015, 2016). The African Great Lakes are thus important regulators of the East African climate, which continues to present a challenge to modellers (James et al., 2018; Woodhams et al., 2018).

Adequately observing and modelling tropical lake-atmosphere interactions often remains a challenge, even though efforts have been made to quantify these exchanges (Verburg and Hecky, 2003; Verburg and Antenucci, 2010; Thiery et al., 2014a, b; Delandmeter et al., 2018; Weyhenmeyer et al., 2017). Moreover, important uncertainties remain present in several of the reference products, notably regarding precipitation (Dinku et al., 2008; Sylla et al., 2013; Awange et al., 2015; Kimani et al., 2017), hence the need for high-quality, in-situ meteorological measurements over the African Great Lakes (Anyah and Semazzi, 2004; Anyah et al., 2006; Anyah and Semazzi, 2009). To help address this need, a state-of-the-art automatic weather station was installed in 2012 on Lake Kivu (AWS Kivu).

Lakes interact with the atmosphere via a variety of processes. Physical lake models developed for use in a meteorological context have thus far concentrated on lake-atmosphere interaction through turbulent and radiative fluxes. The effects of rain on LWST, both directly from thermal perturbation, and indirectly from changing the lake stratification, are little understood or represented to date. Evidence of the significance of rain effects, particularly in the tropics, is beginning to emerge however (Wei et al., 2014). Here, we combine theoretical considerations with analysis of the unique multi-annual dataset from AWS Kivu to estimate the significance of the rainfall effect on LWST in the tropics. As will be shown, the uniformity of the Lake Kivu climate increases its appeal as a location at which to assess this effect.

To summarise the structure of the following sections, in section 2 the study area is described; section 3 discusses the mechanisms of rain effect on LWST; the Kivu data are presented and analysed in section 4; and finally the results are discussed.

## 2 Description of Lake Kivu

The African Great Lakes are of utmost importance for regional economies, as well as being essential to the survival of the local population. As the largest reservoir of freshwater in the tropics, they provide numerous ecosystem services to local communities, such as fishing grounds, drinking water and electricity. Lake Victoria alone directly supports 200 000 fishermen operating

from its shores and sustains the livelihood of more than 30 million people living at its coasts (East African Community, 2011). During the last decades, however, the African Great Lakes experienced fast changes in ecosystem structure and functioning, and their future evolution is a major concern (O'Reilly et al., 2003; Verburg et al., 2003; Verburg and Hecky, 2009; Borges et al., 2015). Moreover, outflow dam management, along with climate variability and change, exert a large influence on the water levels of the largest African Great Lake, Lake Victoria (Vanderkelen et al., 2018a, b).

Lake Kivu (01° 35' S – 02° 30' S; 028° 50' E – 029° 23' E) is situated along the border of Rwanda and the Democratic Republic of Congo, and is one of the seven African Great Lakes (figure 1). The lake has a surface area of 2370 km$^2$, lies 1463 m above sea level and is up to 485 m deep. The outflow is located at the lake's southern tip and forms the Ruzizi river, which flows southwards into Lake Tanganyika. Although the lake is meromictic, the oxic mixolimnion deepens to 60-70 m during the dry season. Below that, the monimolimnion is found rich in nutrients and dissolved gases, in particular carbon dioxide and methane (Degens et al., 1973; Borges et al., 2011; Descy et al., 2012; Morana et al., 2015b, a, 2016). Interestingly, temperature and salinity within the monimolimnion increase with depth due to the input of heat and salts from deep geothermal springs (Degens et al., 1973; Spigel and Coulter, 1996; Schmid et al., 2005). Given its high altitude and location close to the equator, surface water temperatures over Lake Kivu are relatively constant throughout the year.

## 3 Physical effects of rainfall

Rain may have an effect on the lake surface in four ways: (i) evaporative cooling of the near-surface air during precipitation, which induces an additional upward sensible heat flux from the lake towards the atmosphere, (ii) by the direct rain heat flux into the lake, (iii) by mixing the lake surface layer through the flux of kinetic energy, and finally (iv) by convective mixing of the lake surface layer. The first of these ought to be parametrised by atmospheric models, as with related atmospheric effects like the reduction of insolation by cloud cover. The others lie mainly in the lake-modelling domain. Hereafter, we discuss points (i)–(iv) above in some more detail.

### 3.1 Cooling of near-surface air

Raindrops falling into unsaturated air will cool through evaporation. Their passage through the air leads to heat transfer from the air, hence cooling the air. As the rainfall continues, the air will tend to saturation, and both rain and air will approach the air's original wet-bulb temperature. Thermodynamically, further quantification and parametrisation of this process requires consideration of various factors such as the atmospheric moisture and temperature profile, drop-size distribution, drop concentration etc. (Schlesinger and Oh, 1988; Feingold, 1993).

At the extreme end of intense convection, atmospheric cooling and momentum transfer from rain may produce cold convective downdraughts, which transport cold air to the surface from higher levels (Kamburova and Ludlam, 1968; Knupp and Cotton, 1985; Srivastava, 1987). These additional dynamic effects are potentially less well represented or resolved in atmospheric models, but are of importance in terms of gust hazard, as well as for their relevance to modelling of the convective cycle (Rooney, 2015; Thiery et al., 2017).

## 3.2 Direct heat flux

The specific heat capacity of water is approximately $4.2 \times 10^3$ J kg$^{-1}$ K$^{-1}$. A rainfall rate of 1 mm hr$^{-1}$ is thus equivalent to a heat flux of approximately $1.2 \Delta T$ W m$^{-2}$, where $\Delta T$ is the temperature difference between the rain and the surface which absorbs it. Rain temperature variation relative to air or surface temperature is not well-known. According to Byers et al. (1949), rain may be much colder than the ambient air at the start of a thunderstorm, but possibly comparable at later stages. This is presumably due to evaporative cooling reducing the air temperature to nearer that of the rain over time, and is consistent with the approximation of the rain temperature to the wet-bulb temperature (Wei et al., 2014, see also references therein).

On seasonal to decadal timescales, the sensible heat contribution by rainfall is deemed small (Verburg et al., 2011). Wei et al. (2014) have estimated that this flux is largest in the tropics, with seasonal mean values of the order of -2 W m$^{-2}$, and they also state that neglect of this contribution may partly explain some air temperature biases in climate reanalyses. Van Beek et al. (2012) have noted its significance for correctly estimating the surface temperature of tropical rivers. On much shorter timescales, surface cooling due to rain may affect weather patterns. On land, this process has been widely studied (Seneviratne et al., 2010; Taylor et al., 2012; Guillod et al., 2015; Lorenz et al., 2016). The rain effect in water bodies is more uncertain, although recently there have been some interesting observational studies e.g. Reverdin et al. (2012).

## 3.3 Mechanical and convective mixing

As well as the direct effects of an additional heat flux, rainfall may produce a perturbation of LWST by the mechanical and convective mixing of the near-surface portion of the lake.

An early study by Katsaros and Buettner (1969) indicated that large drops (3 mm diameter) produce mixing to depths of 10 cm and smaller drops ($\leq$1.2 mm) mixed to perhaps one third of this depth. Similarly, Green and Houk (1979) found that the presence of some large drops is very important in producing significant subsurface mixing.

Several subsequent studies of artificial rainfall have examined rainfall-generated turbulence in slightly more detail. Artifical, heavy rainfall has been observed to produce turbulent mixing over depths of 10–20 cm in the study of Lange et al. (2000), with a drop size approximately 3 mm, and that of Zappa et al. (2009) with a distribution of drop sizes in the range 0.3–5.3 mm which was modelled on the measured natural-rain distributions of Marshall and Palmer (1948). By contrast, Harrison and Veron (2017), with drop sizes of approximately 1.3 mm, found turbulent kinetic energy (TKE) to be independent of artificial rainfall intensity, at even higher intensities, and suggested that a significant fraction of the energy went into producing small-lengthscale or capillary motions within 1–2 cm of the surface. Again, these results would seem to indicate that drop size is an important factor. While the effects of drop impact in a deep liquid reservoir are quite complex, even for a single drop (Prosperetti et al., 1989; Rein, 1996), it may be reasonable as a first hypothesis to assume that a significant fraction of the kinetic energy of natural heavy rain goes into subsurface TKE production.

The kinetic energy flux of real rainfall has also been estimated in the context of soil erosion studies (Salles et al., 2002; van Dijk et al., 2002; Yu et al., 2012). A rain rate of 5 mm h$^{-1}$ may give rise to a kinetic energy flux $F_K$ of around 0.02 W m$^{-2}$, for example. The rate of TKE production in the surface boundary layer of a lake may then be assumed to scale as $\phi F_K/(\rho_w \ell)$

where $\ell$ is the depth to which the rain penetrates, $\rho_w$ is the water density and $\phi F_K$ is some fraction of $F_K$ where $\phi \leq 1$ (see for example Townsend (1976, Ch.2) for a discussion of TKE evolution). The significance of this rain-driven production of TKE may be gauged by comparison with the kinetic energy input from wind shear.

Mechanical surface forcing by wind upon lakes is usually modelled through matching of stress, so that the aqueous friction velocity at the lake surface $u_{*l}$ is

$$u_{*l} = \left( \frac{\rho_a}{\rho_w} \right)^{1/2} u_* \tag{1}$$

where $u_*$ is the friction velocity in the atmospheric surface layer, and $\rho_a$ is the air density. A typical atmospheric friction velocity of order 1 m s$^{-1}$ then implies an aqueous value of $u_{*l} \approx 0.03$ m s$^{-1}$ (Anctil and Donelan, 1996; Csanady, 2001). The lake surface TKE production from wind-driven shear scales as $u_{*l}^3/(\kappa z)$ where $z$ is the depth and $\kappa = 0.4$ is von Karman's constant (e.g. Skyllingstad and Denbo, 1995).

Thus, setting $z = \ell$, the turbulent mixing from rain may be compared to that from wind shear by comparing $u_{*l}^3/(\kappa \ell)$ with $\phi F_K/(\rho_w \ell)$, or equivalently by comparing $u_{*l}$ with $(\kappa \phi F_K/\rho_w)^{1/3}$. This last term is evaluated as of order 0.02 m s$^{-1}$ for heavy rain, using the value of $F_K$ given above and $\phi$ in the range 0.5–1, and is the same order of magnitude as the friction velocity for a moderate-to-strong wind. Hence, the turbulent mixing rates due to wind-shear and rain may at times be comparable in the top few centimetres of a lake.

For cold rain falling onto a relatively warm lake, convective effects will presumably add to the mixing strength and depth. While Green and Houk (1979) concentrated mainly on the case of warm rain falling onto cold water, their experiments with the opposite temperature contrast showed cooling throughout the depth of their reservoir i.e. to at least 0.4 m. The LWST perturbation caused by mixing effects will depend on the stratification of the lake near-surface region.

## 4 Data analysis

### 4.1 Instrumentation and measurements

AWS Kivu is installed on the research platform of the Rwanda Energy Company, approximately 3 km offshore of the cities of Gisenyi (Rwanda) and Goma (D.R. Congo, see figure 2). Since 9 October 2012, AWS Kivu has provided continuous, high-quality observations of near-surface meteorology and four-component radiation. The continuous time series obtained so far from AWS Kivu is, to our knowledge, unique of its kind in the tropics.

AWS Kivu consists of sensors for air temperature ($T_a$, C), relative humidity ($RH$, %), air pressure ($p$, Pa), precipitation ($P$, mm), wind speed ($U$, m s$^{-1}$) and direction ($WD$, °) and the four radiation components ($SW_{in}$, $SW_{out}$, $LW_{in}$, $LW_{out}$, all in W m$^{-2}$). Details of the sensors are given in table 1. LWST is calculated from the upwelling longwave irradiance using the Stefan-Boltzmann law and assuming an emissivity of 0.99 (Wan, 2008). The station is powered by a solar panel, and all sensors are placed at a height of 4.40 m above the water surface (a in figure 2), except for wind speed and direction which are measured at 7.20 m above the lake (b in figure 2). While the station is mounted on a metal container (c in figure 2), efforts were made to

**Table 1.** AWS Kivu sensor specifications.

| Data | Sensor | Range | Accuracy |
|------|--------|-------|----------|
| Air pressure | Cambell Scientific CS100 | 600 – 1100 hPa | 0.5 hPa |
| Air temperature | Cambell Scientific CS215 | -40 – +70 C | 0.3 K |
| Relative humidity | Cambell Scientific CS215 | 0 – 100 % | 2 – 4 % |
| Wind speed | Young 05103 | 0 – 100 m s$^{-1}$ | 0.3 m s$^{-1}$ |
| Wind direction | Young 05103 | 0 – 360° | 3° |
| Precipitation | Tipping Bucket ARG100 | 0.2 – 500 mm hr$^{-1}$ | 95 – 98 % |
| SW components | Kipp and Zonen CNR4 | 305 – 2800 nm | <5 % |
| LW components | Kipp and Zonen CNR4 | 4500 – 42000 nm | <10 % |

minimise its effect on the meteorological measurements. Notably, temperature, humidity and radiation sensors were mounted 6 m horizontally away from the container edge, making sure that recorded conditions are representative for the water surface.

Variables are sampled every 15 seconds, from which 30 minute averages are calculated and stored. In the case of precipitation, accumulated values are stored, and for wind speed both mean and maximum values are recorded. Moreover, short periods of high-frequency radiation measurements enable an assessment of the potential effect of platform movements. Through a General Packet Radio Service (GPRS), the KU Leuven Regional Climate Studies group receives the observations directly from the station, allowing for remote problem detection.

The time span of measurements used here was between 13 September 2012 and 14 August 2017, however most of the analysis is based on four calendar years of data from 1 January 2013 to 30 December 2016.

## 4.2 Weather and climate

The data indicate that there is a remarkable uniformity of the lake climate. The annual air temperature range is around 14 K. The daily rainfall totals for 4 calendar years show a generally uniform spread, but with a slightly drier period around July (figure 3). There seem to be two prevailing wind directions which do not vary greatly with the weather, inasmuch as this is represented by rainfall (see section 4.3). This uniformity is beneficial for the following analysis, since there are consequently fewer sources of variation upon which the lake behaviour may potentially depend.

The temporal variations in weather may be examined further using power spectra of rainfall and wind speed (figure 4). These show that a large part of the variation is on the diurnal scale. This provides justification for the use of average diurnal cycles to explore the behaviour of the system.

We also note the presence of some sub-daily peaks in the wind speed spectrum, the most dominant at a frequency corresponding to a period of approximately 8 hours, and the next two corresponding to periods of approximately 6 hours and 12 hours. (As will be shown later, the sub-daily wind fluctuations giving rise to these peaks are evident on plots of mean daily

wind speed. These fluctuations are probably due to local circulations caused by lake or land breezes, and the largely bi-modal distribution of the wind direction, also shown later, appears to support this interpretation.)

## 4.3 Partitioning by rainfall

To examine the effect of heavy rain, four years of data will be analysed (1 January 2013 to 30 December 2016). This amounts to 1457 days, as three days are omitted due to missing data. These data are referred to as ALL data in the following analysis. Based on daily rainfall totals, they may be divided into DRY, WET and VWET days. DRY days are days with no rainfall. The remaining days are partitioned into WET or VWET depending on whether the rainfall total is respectively less or greater than a threshold of 8 mm (figure 3). The number of days of each type is DRY: 690, WET: 585, VWET: 182.

Regarding the distribution of hourly rainfall over the four years, 2.4% of hours had a rainfall total greater than 1 mm, and 0.6% of hours had a rainfall total greater than 5 mm. Figure 5 shows the average hourly rainfall on WET and VWET days. Both show a minimum in rainfall around 06 UTC (08 LT, LT = UTC + 2 h), and a peak around the middle of the day. There is also a later precipitation maximum in VWET, which may indicate the development of nighttime heavy storms.

The effect of daily weather on LWST is summarised in figure 6. It can be seen that the average diurnal cycles of air temperature and relative humidity are quite smooth, with a spread related to the rainfall category. Thus DRY days are the warmest and least humid, VWET days are the coldest and most humid, and WET and ALL days lie between these extremes. Both the air temperature and LWST start close together for the WET and VWET categories, but by the end of the day there is a difference in the mean, with VWET being colder than WET. Specifically, the average temperature difference over the last 6 hours of the day is 0.42 K.

Atmospheric forcing of LWST is usually characterised in terms of turbulent or radiative fluxes, with turbulent fluxes depending on mean wind speed, lake-air temperature difference and near-surface humidity. For the categories described here, the choice of partitioning threshold between WET and VWET days coincidentally produces extremely similar graphs of mean wind speed. This has the effect of removing an important potential source of variation between these categories. The distributions of wind directions and speeds are also quite uniform, see figures 7 and 8.

Figure 9 shows the difference between the WET and VWET cases in terms of mean net radiation, on average over the course of a day. The lake has absorbed approximately $1.7 \times 10^6$ J m$^{-2}$ more in the WET case. This is due presumably to WET days having less (or less thick) cloud cover than VWET days on average. This source of difference may be eliminated by adding a further constraint to the total absorbed radiation on WET days. This is described in the next subsection.

## 4.4 Further partitioning by rainfall and net radiation

WET days have a higher mean net radiation than VWET days (figure 9). To separate the effects of rainfall and radiation, the WET days may be further filtered for those with integrated net radiation below a specified amount, so that the mean value is reduced to equal to, or less than, that on VWET days. This subset of "dull" WET days is labelled DWET. With a threshold total radiation of $1.5 \times 10^7$ J m$^{-2}$, the average total radiation of DWET days is approximately $1.03 \times 10^7$ J m$^{-2}$, compared to an average total radiation of $1.05 \times 10^7$ J m$^{-2}$ on VWET days. It is worth emphasising that the radiation threshold for DWET days

was deliberately chosen to allow a small margin, so that VWET days absorbed 1–2% more radiation in the mean than DWET days. Thus any extra LWST cooling in VWET days compared to DWET days would be against a background of a slight excess of absorbed radiative energy on VWET days, in the mean.

Using this additional constraint, the number of DWET days in the 4-year period is 425, or 73% of the WET days. Table 2 summarises the number of days of each type in each year, along with data on which type of day came immediately before and after. The distributions of wind directions and speeds for the DWET category are also plotted in figures 7 and 8. The diurnal cycle of the mean net radiation difference between DWET and VWET cases is shown in figure 9.

The average daily rainfall on DWET days is 2.31 mm, compared to 2.33 mm on WET days and 17.99 mm on VWET days. Thus, the contrast in rainfall amount is largely preserved by this resampling. The diurnal evolution is also plotted in figure 5, again showing that DWET is similar to WET.

The average diurnal evolution for the categories of ALL, DRY, DWET and VWET is shown in figure 10. It can be seen that the evolution of diurnal wind speed is reasonably unchanged for DWET days compared to that of WET days (figure 6), but the evolution of LWST on DWET days is closer to that of VWET days. However, most of the difference in surface temperature between these categories in the last few hours of the day remains, with an average temperature difference over the last 6 hours of 0.29 K.

Considering the reliability of this difference, it may be noted that a difference of 0.3 K against a background at approximately 300 K is equivalent to a difference in upwelling longwave of approximately 1.8 W m$^{-2}$. However, since this is a difference between mean values taken over a minimum of 182 observations at each time of day, it should be compared with the standard error of the mean. That is, it only requires instrumental accuracy of $1.8 \times \sqrt{182} \approx 24$ W m$^{-2}$, which is within the instrument specification (table 4.1).

In terms of significance, the standard error of the difference between the mean DWET and VWET values is given by

$$SED = \sqrt{\frac{\sigma^2_{DWET}}{N_{DWET}} + \frac{\sigma^2_{VWET}}{N_{VWET}}} \qquad (2)$$

where $\sigma^2_{DWET}$ and $\sigma^2_{VWET}$ are the variances of LWST in the DWET and VWET cases respectively, and the number of observations at any particular time of day are $N_{DWET} = 425$ and $N_{VWET} = 182$, as stated earlier. $SED$ takes values in the range 0.05–0.07 K during the first and last few hours of the day, climbing to over 0.14 K during daylight hours. The hypothesis that the means are equal may be tested using the difference in mean values divided by $SED$ (e.g. Frank and Althoen, 1994, chapter 10). This is plotted in figure 11. It can be seen that, before 16 UTC, this statistic takes values in the approximate range $[-2, 2]$, indicating that the hypothesis of equal means may be accepted at approximately the 5% level of significance at these times. However, after 17 UTC, this statistic climbs to values well above 2, indicating that the hypothesis may reasonably be rejected at at later times. It is therefore concluded that the difference in the means during the last few hours of the day is statistically significant.

Finally, the effect of rain on the sensing of LWST should also be considered as a possible cause of observed LWST differences. From an atmospheric modelling viewpoint, the sensed surface temperature is the important quantity in many cases, as has been recently discussed in the context of the introduction of a "skin" temperature into the FLake lake model (Le Moigne et al., 2016). In this particular case, it may be remarked that while the rain rate during VWET days is highest during the final

**Table 2.** Percentages of the types of day which came immediately before and after each type, broken down by observation year. N denotes the number of days of each type in each year. OTH denotes days classified as WET but not DWET, so that the OTH and DWET categories combine to make up the WET category. The data from each year are presented separately to give an indication of inter-annual variation. To give an example from the last line of the table, in 2016 there were 45 VWET days, and 27% of the days in 2016 directly preceding VWET days were DRY, compared to 18% of the days directly following VWET days.

| | % preceding days | | | | N | % following days | | | |
|---|---|---|---|---|---|---|---|---|---|
| | DRY | OTH | DWET | VWET | | DRY | OTH | DWET | VWET |
| **2013** | | | | | | | | | |
| DRY | 66. | 5. | 21. | 7. | 183 | 66. | 7. | 16. | 11. |
| OTH | 44. | 22. | 26. | 7. | 27 | 37. | 22. | 22. | 19. |
| DWET | 28. | 6. | 46. | 20. | 109 | 36. | 6. | 46. | 12. |
| VWET | 43. | 11. | 28. | 17. | 46 | 29. | 4. | 49. | 18. |
| **2014** | | | | | | | | | |
| DRY | 68. | 7. | 20. | 6. | 167 | 68. | 8. | 18. | 6. |
| OTH | 31. | 17. | 40. | 12. | 42 | 26. | 17. | 33. | 24. |
| DWET | 29. | 13. | 37. | 21. | 106 | 31. | 16. | 37. | 16. |
| VWET | 20. | 20. | 34. | 26. | 50 | 20. | 10. | 44. | 26. |
| **2015** | | | | | | | | | |
| DRY | 66. | 6. | 23. | 5. | 158 | 66. | 5. | 20. | 9. |
| OTH | 19. | 21. | 48. | 12. | 42 | 24. | 21. | 38. | 17. |
| DWET | 26. | 13. | 44. | 18. | 124 | 29. | 16. | 44. | 11. |
| VWET | 34. | 17. | 34. | 15. | 41 | 20. | 12. | 54. | 15. |
| **2016** | | | | | | | | | |
| DRY | 68. | 12. | 16. | 4. | 182 | 68. | 9. | 17. | 7. |
| OTH | 33. | 14. | 27. | 27. | 49 | 43. | 14. | 22. | 20. |
| DWET | 35. | 13. | 35. | 16. | 86 | 35. | 15. | 35. | 15. |
| VWET | 27. | 22. | 29. | 22. | 45 | 18. | 29. | 31. | 22. |

Note that calculating percentages to the nearest percent occasionally produces sets that do not sum to 100 exactly.

few hours of the day, figure 5 shows that the average VWET rain rate is also appreciably higher than that of the DWET category at some earlier times. However the LWST differences, and their significance, at these earlier times are both much less than during the final few hours of the day, as shown in figures 10 and 11. This provides some evidence that a systematic effect of rain on the sensor is not the main cause of LWST differences.

## 5   Discussion and Conclusion

Lake Kivu has a remarkably stable tropical lake climate, and AWS Kivu has yielded a high-quality, multi-year, over-lake observational record which is rare and perhaps unique in the tropics, and well-suited to the present research question. This study is the first such use of these data.

Data over four years from AWS Kivu have been categorised by daily rainfall amount and net radiation, to investigate the possible effects of rainfall on lake water surface temperature (LWST), which may be particularly significant in the tropics (Wei et al., 2014). The choice of division between days with heavy or light-to-moderate rain (respectively greater or less than 8 mm total) has helped minimise the sources of difference between the categories other than that due to rainfall. Spectral analyses have shown that, in this uniform climate, one of the dominant variations is the diurnal cycle, and hence the different categories are compared via their mean diurnal evolution. In the mean data examined here, heavy rain on a tropical lake would seem to have the capability to produce a reduction of a few tenths of one Kelvin in LWST over the course of several hours at the end of the day, compared to light-to-moderate rain; and this reduction is statistically significant.

The possible pathways by which this effect may arise are: (i) cooling of the air due to contact with evaporatively-cooled raindrops, and a subsequent increase in atmospheric sensible heat flux from the lake, (ii) negative heat flux directly to the lake from rain impingement, (iii) mechanical mixing of the lake surface layer by the kinetic energy of rain impact, and (iv) convective mixing of the lake surface layer due to the negative heat flux from rain. Of these, the first is the most likely to be a parametrised process in a General Circulation Model of the atmosphere, although it could be considered the most indirect of the four. The rain heat flux is likely to be proportional to the difference between the air wet-bulb temperature and LWST. We have indicated with scaling arguments that the mechanical mixing due to heavy rain may be comparable to that of a strong wind. The convective mixing will depend on the near-surface temperature structure of the lake, and hence on its recent history.

Unfortunately, the available data do not cover several other process-related quantities that would be useful to have, such as turbulent heat fluxes, rain temperature, fine-scale lake temperature profiles or lake turbulence measurements. Thus, the processes producing this effect are not directly measured. However, through our indirect analysis of the processes it seems likely that cooling by rain combined with mechanical and convective mixing from droplet impact may have an effect on LWST, in addition to the effect from the more widely studied pathway of evaporative cooling.

Potential avenues of future work would be to examine these processes more closely in a targeted campaign of observations, including the quantities listed above, and to consider how lake models may be modified to include their representation. An intermediate step in the latter might be to re-examine previous modelling studies to explore correlations between lake-model errors and rainfall records. Since, as discussed earlier, rainfall may affect not only the surface temperature but potentially also the temperature or depth of any upper mixed layer, some or all these quantities could be susceptible to rainfall effects. For models that predict vertical fluxes through the water column, comparison of these with any available flux or TKE measurements would be a possible way to estimate rain penetration in real lakes. There is an indication in the data of Reverdin et al. (2012) that rainfall effects may have a sudden onset but a subsequent slower decay, so that some filtering method such as an exponential

moving average applied to the rainfall data may be required when considering correlations. The decay timescales of any such filtering could also have a depth dependence.

In large tropical lakes, it is possible that a surface temperature difference of order half a Kelvin may suppress or enhance the strength of local air circulations, such as lake breezes, and hence have some effect (or even feedback) on the evolution of the local weather (Thiery et al., 2015, 2016). For example, the length of time between severe storms may be partly affected by the recovery timescale of LWST. In the short term it would seem possible to incorporate, perhaps semi-empirically, the effect of rain temperature and rain-induced turbulence into simple lake models as used for weather and climate modelling.

**Data availability:** The data are available on request from the dataset owners, Wim Thiery and Nicole van Lipzig.

*Acknowledgements.* WT was supported by an ETH Zurich postdoctoral fellowship (Fel-45 15-1). The Uniscientia Foundation and the ETH Zurich Foundation are thanked for their support to this research. The Belgian Science Policy Office (BELSPO) is acknowledged for the support through the research project EAGLES (CD/AR/02A). We thank Stijn Bruggen, who analysed the AWS data in his Master's thesis, and thereby supported the design of this study and the analysis presented here. GGR thanks John M. Edwards for a helpful discussion of TKE budgets.

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

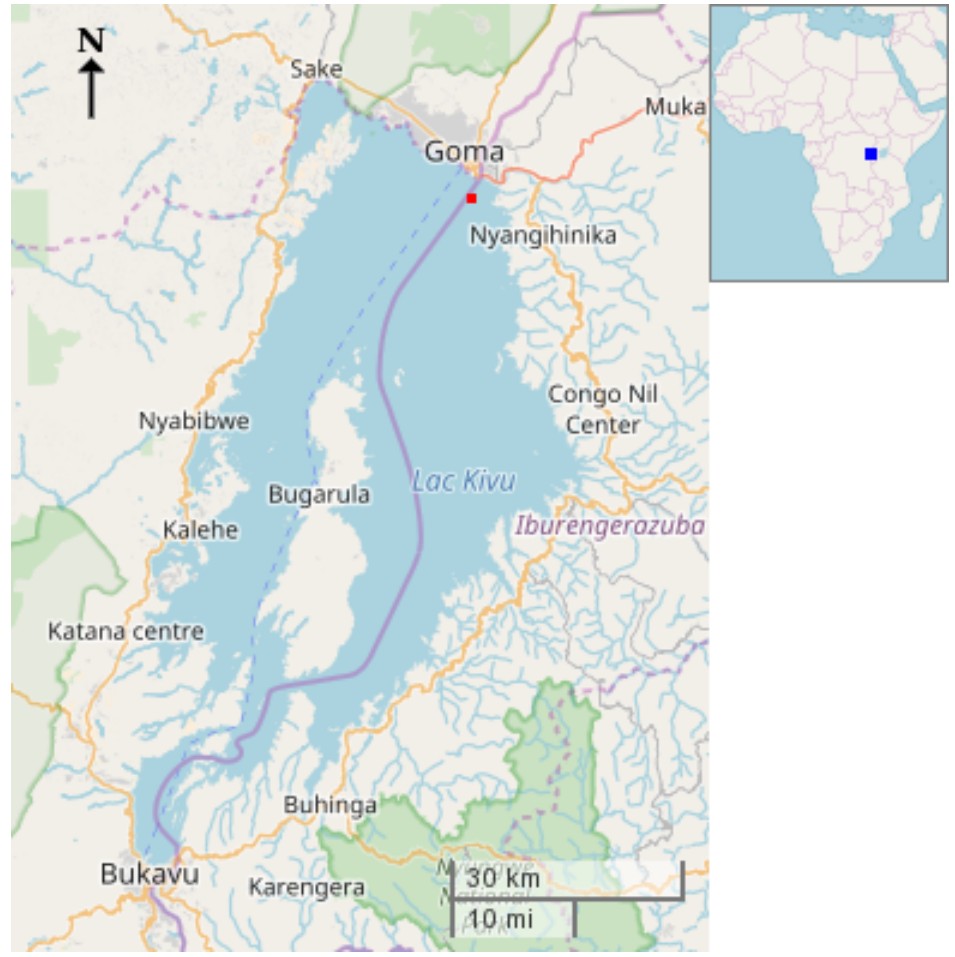

**Figure 1.** Maps of Lake Kivu geography and situation. The lower left corner of the large map is at 2.6 S, 28.7 E, and the upper right corner is at 1.5 S, 29.5 E. The lake is approximately 90 km long and 50 km wide. The magenta line along the lake indicates the boundary between D.R.Congo to the west and Rwanda to the east. The weather station is situated approximately 3 km offshore, near Goma at the northern end of the lake. Its position is marked with a red square. The location of Lake Kivu within Africa is marked on the small map by a blue square. (©OpenStreetMap contributors. Mapping data are available under the Open Database License.)

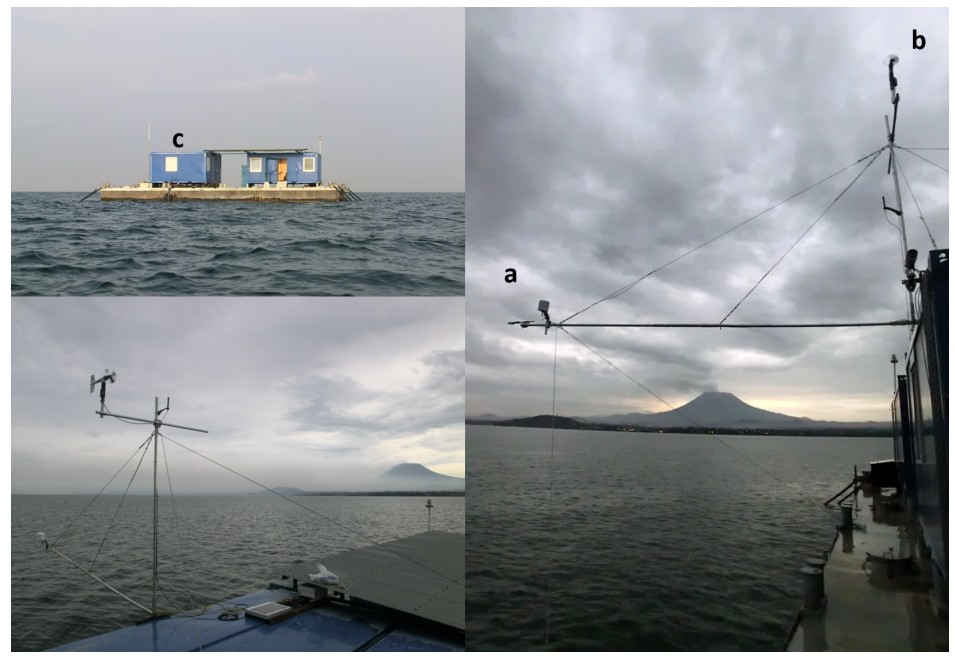

**Figure 2.** Automatic weather station on Lake Kivu after its installation, 8 October 2012 (©Wim Thiery). **a** indicates the location of the temperature, relative humidity and radiation sensors at 4.40m above the lake surface. **b** shows the location of the wind vane at 7.20 m above the lake surface. **c** indicates the container on which the station was mounted.

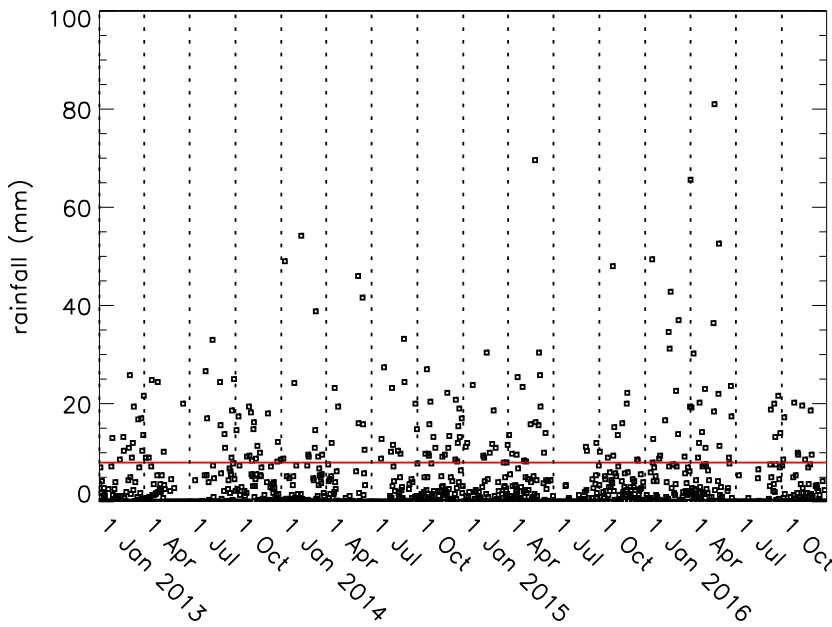

**Figure 3.** Daily rainfall totals for four years of the observational record, beginning on 1 January 2013. The red line marks the 8 mm point, which is used to partition rain days between WET ($\leq$ 8 mm) and VWET($>$ 8 mm).

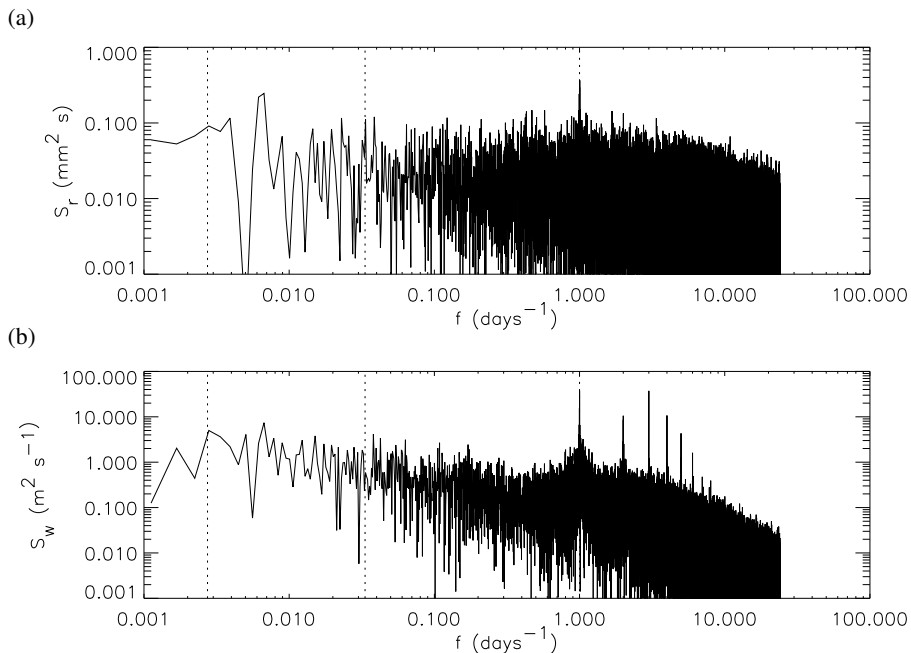

**Figure 4.** (a) Power spectrum of half-hourly rainfall amount ($S_r$, mm$^2$ s), and (b) power spectrum of half-hourly mean wind speed ($S_w$, m$^2$ s$^{-1}$), both for the period 13 September 2012 to 14 August 2017. The vertical dotted lines mark frequencies ($f$, days$^{-1}$) corresponding to 1 day, 30 days and 365 days. Both plots show a distinct peak at the daily frequency, with the wind speed also exhibiting several sub-daily peaks.

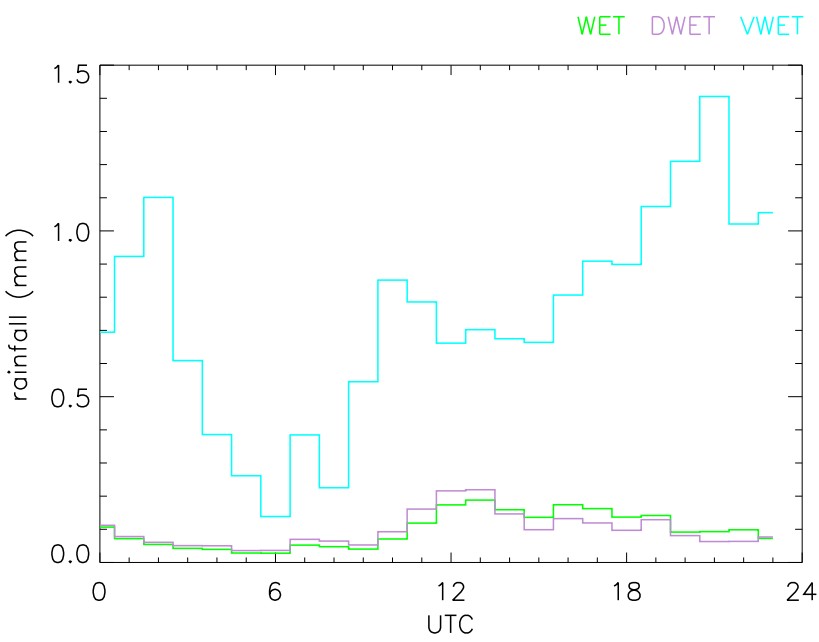

**Figure 5.** The average rainfall for each hour of the day, for VWET days (cyan) WET days (green) and DWET days (magenta). On this and later plots, time is shown as UTC (Universal Time Coordinate), which is 2 hours behind LT (Local Time) i.e. LT = UTC + 2 h.

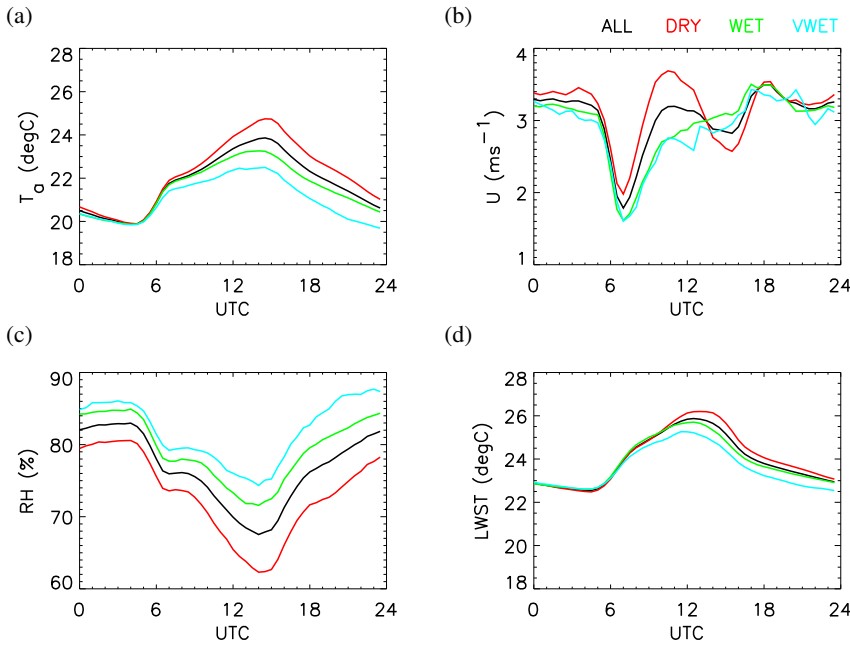

**Figure 6.** Mean diurnal cycles of (a) air temperature ($T_a$, C), (b) wind speed ($U$, m s$^{-1}$), (c) relative humidity ($RH$, %), (d) lake surface temperature (LWST, C) from the upwelling longwave irradiance, assuming an emissivity of 0.99. The input data are those from the four years 2013-2016 of the campaign, to represent all seasons equally. The colours correspond to diurnal cycles averaged over ALL days (black), DRY days (red), WET days (green) and VWET days (cyan).

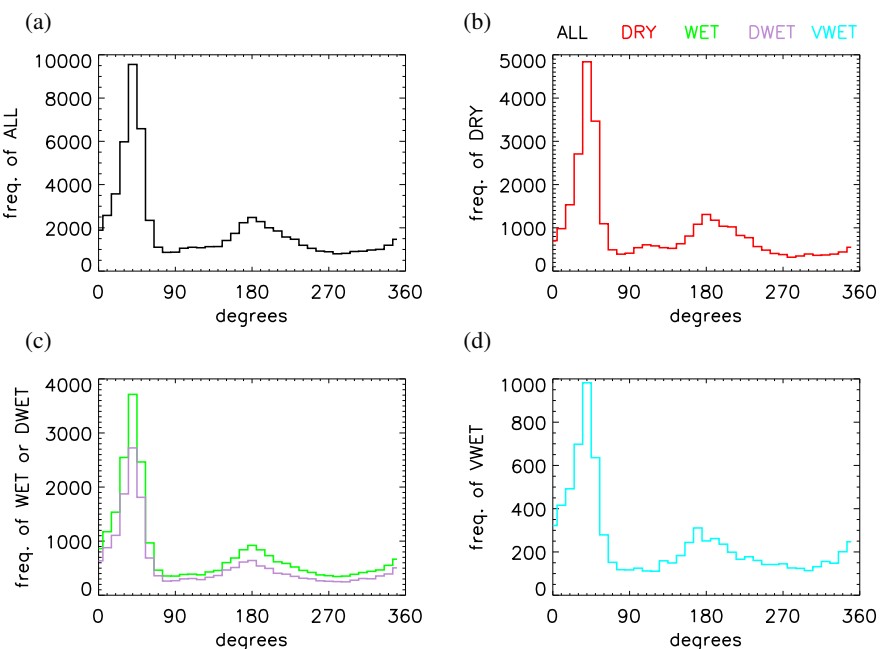

**Figure 7.** Histograms of half-hourly mean wind direction for 2013–2016. The colours are the same as in figures 5 and 6.

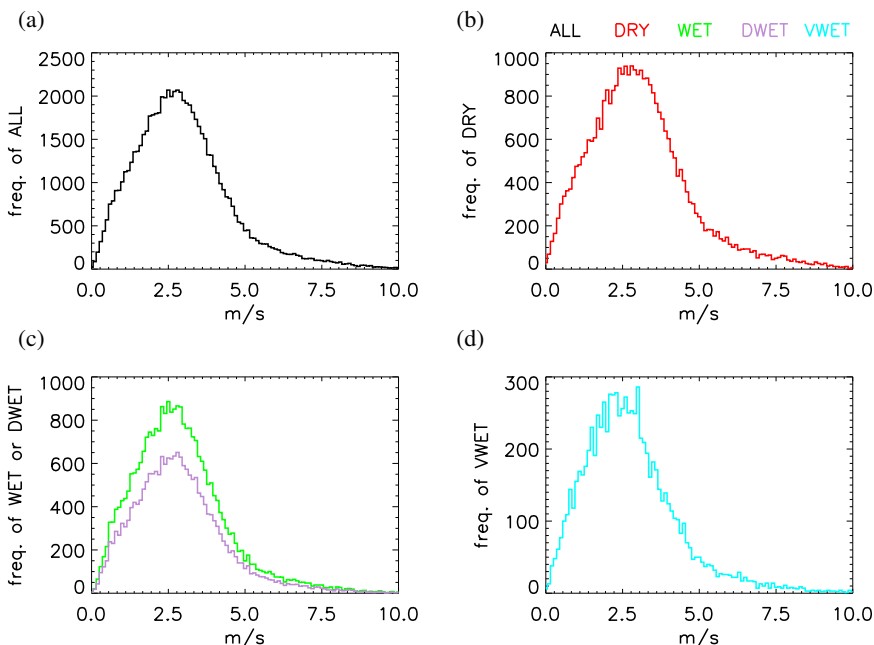

**Figure 8.** Histograms of half-hourly mean wind speed for 2013–2016. The colours are the same as in figures 5 and 6.

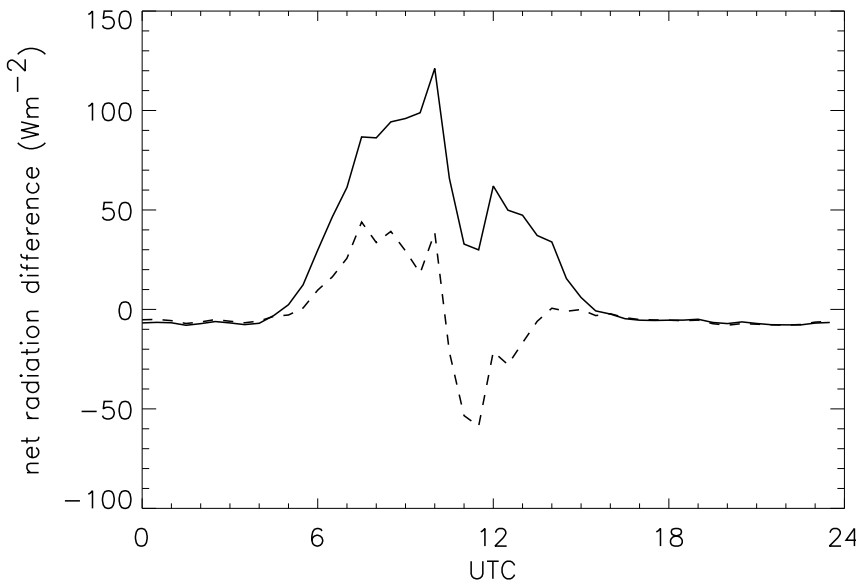

**Figure 9.** Comparison of the differences in net radiation. WET minus VWET net radiation is the solid line, and DWET minus VWET net radiation is the dashed line.

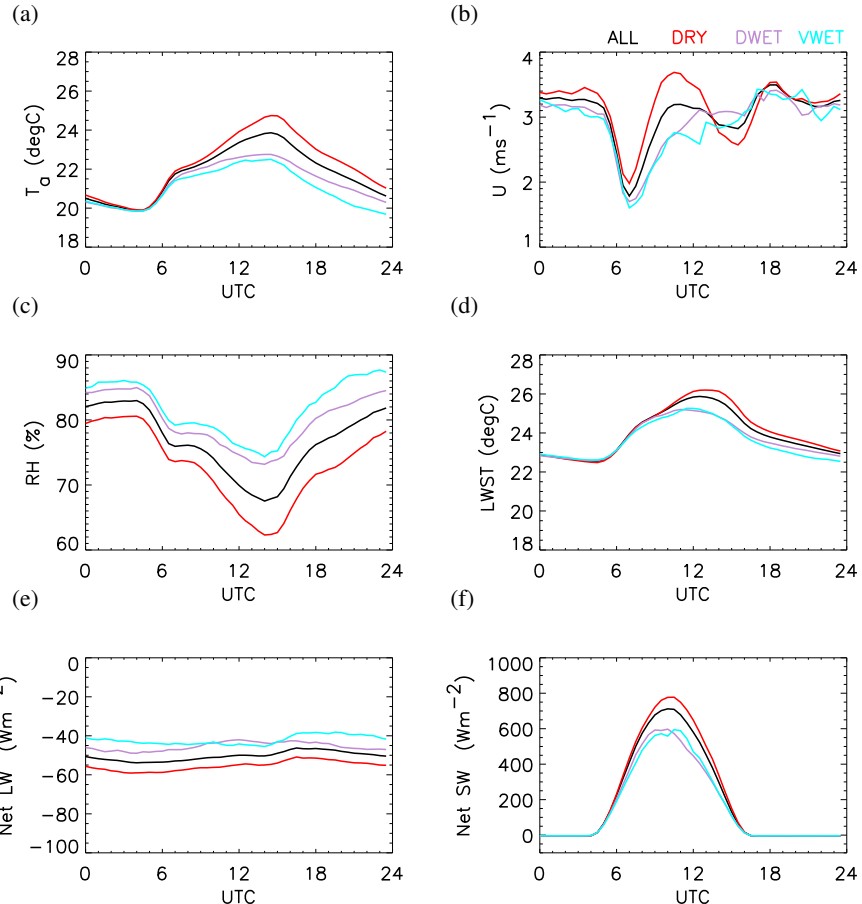

**Figure 10.** Mean diurnal cycles of (a) air temperature ($T_a$, C), (b) wind speed ($U$, m s$^{-1}$), (c) relative humidity ($RH$, %), (d) lake surface temperature (LWST, C) from the upwelling longwave irradiance, assuming an emissivity of 0.99, (e) net longwave irradiance (Net LW, Wm$^{-2}$), (f) net shortwave irradiance (Net SW, Wm$^{-2}$). The input data are those from the four years 2013-2016 of the campaign, to represent all seasons equally. The colours correspond to diurnal cycles averaged over ALL days (black), DRY days (red), DWET days (magenta) and VWET days (cyan).

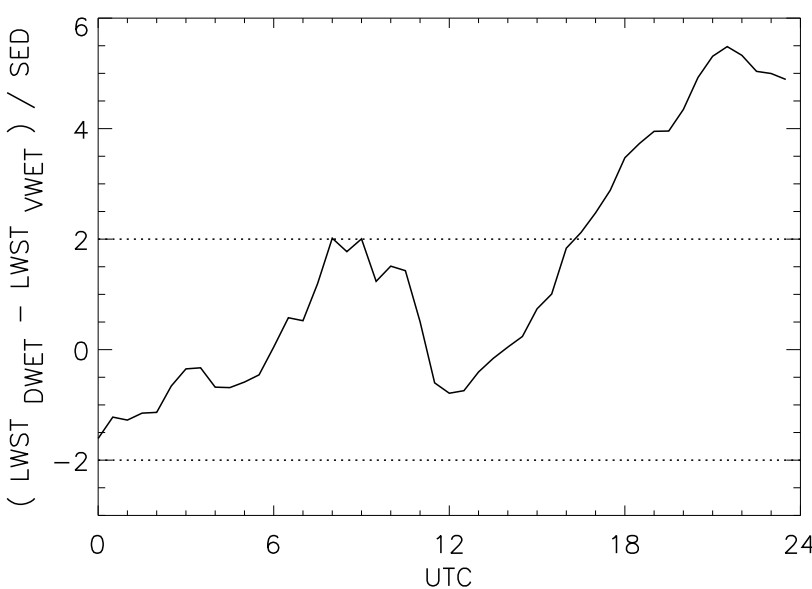

**Figure 11.** Diurnal behaviour of the difference in mean values of LWST in the DWET and VWET cases, normalised by the standard error of the difference, $SED$, see (2). The dotted lines show $\pm 2$ standard devations for this statistic. It may be seen that the value climbs above 2 at later times, indicating the significance of the difference in the means then.