# Peer review of "Estimating the effect of rainfall on the surface temperature of a tropical lake"

_Hydrology and Earth System Sciences, 2018_

## Referee Comment (RC1) · Anonymous Referee #1 · 25 Sep 2018

GENERAL COMMENTS

The authors propose an interesting and quite plausible argument, both through data and scaling analyses, for a small cooling effect on the surface temperature of a tropical lake due to heavy rainfall. In the lake under consideration a cooling of 0.3 K is proposed under heavy rainfall conditions, small but nevertheless likely climatologically significant. The manuscript is concise, thoughtful, and well written, but would benefit from a little more clarification on a few points as outlined below.

SPECIFIC COMMENTS

The authors propose four possible mechanisms for the observed cooling. The first – near surface evaporative cooling of raindrops – is noted as a purely atmospheric phe-

nomenon, and thus excluded from further discussion as the primary interest seems to be lake model parameterizations (Pg. 3, lines 18-19). This exclusion seems a bit hasty: there is no indication in the title emphasizing lake modelling, and the abstract explicitly states the importance for atmospheric models (Pg. 1, lines 10-12). A little more discussion of this mechanism is warranted or the target audience should be identified as lake modelers in order to remove this slight inconsistency.

The scaling analysis for mechanical mixing is interesting, but there is an underlying assumption that all of the incoming mechanical energy (i.e. the kinetic energy of the falling rain drops) is converted into TKE available for mixing. Is this process really 100% efficient? I expect some of the energy must go nearly instantaneously into heat through viscosity. A slightly more sophisticated scaling argument might be able to show that this fraction of energy is very small. At least the possibility should be mentioned.

Radiometers have been used for measuring lake surface temperatures for many years, but there may be slightly different systematic errors under different conditions (i.e. heavy rain vs dull day). Since the temperature effect is relatively small, this might be important and worth a mention. There may be literature on this subject. Any comparison of the radiometric temperatures measured at this site with actual in situ instruments would be helpful – but I gather there are no in situ data? Note that the authors have addressed random measurement errors but not the possibility of different systematic errors.

Fig. 9 shows that during daytime the net radiation from WET exceeds VMET by quite a lot, presumably due to more SW radiation reaching the surface. This is the justification for introducing the DWET category. Fig. 9 also shows that during the night (16 – 5 UTC) the net radiation (i.e. net LW) for VWET exceeds both that for WET and DWET (by 7 or 8 W/m2) – implying either a colder surface or increased downward LW for VWET. This period largely coincides with the period of greatest rainrates for VWET shown in Fig. 5. It seems to me this strengthens the authors' argument that heavy rainfall cools the surface, but they don't point this out. This may be worth mentioning. Fig. 11 also

shows the strongest signal is during 16 – 24 UTC.

MINOR COMMENTS AND FIGURES

Page 3, line 31: This claim needs a reference.

Page 8, lines 16-18: Not clear why you have ruled out mechanism 1 (evaporative cooling)

Fig. 1: would benefit from a context map inset showing location of the lake within Africa

Fig. 2: what are the large boxes in the upper left-hand image?

Fig. 10: there seems to be a large jump between 23 UTC and 0 UTC. This does not seem physically possible, especially where the curves are changing smoothly. Is there a processing error?

---

## Referee Comment (RC2) · Anonymous Referee #2 · 2 Oct 2018

The authors explore a unique long-term observational dataset on a tropical lake to evaluate the role of rainfall on lake surface temperature. In addition to the data analysis, the authors present several scale analysis regarding the direct heat flux and mechanical and convective mixing. The authors found a reduction of lake temperature of about 0.3k in days of heavy rainfall when compared with days with light-to-moderate rainfall, suggesting that further efforts in quantifying and representing this process is important in other regions as well as to be included in atmospheric models. The manuscript is well organized and written presenting an new diagnostic of a process that has not been much explored. Despite the novelty of the work and data used, I have some concerns regarding the potential influence of the radiative fluxes on the diagnostics. Furthermore, a modeling study complementing the observational analysis would strongly enrich the study. Therefore, I recommend the manuscript to be accepted after the authors have addressed some following comments:

Section 3, ln 15: The authors point 4 processes in which rainfall can affect lake surface temperature. Since the authors mention the evaporative cooling, the solar radiation shading during daytime associated with clouds could be also mentioned as a process which should be, in principle properly represented by the atmospheric model

Figure 4: Power spectrum of wind: There are several peaks on the sub-daily frequencies. Could the authors provide the frequencies of these and comment on their source (breeze effects?)

The authors filtered the effect of radiation by defining the DWET days as days with net radiation below 1.5x10**7 J m-2. The average different between DWET net radiation and VWET is about -2.3 W m-2. Visual inspection of T and LSWT mean diurnal cycles for VWET suggests a temperature difference between air and LSWT of about 22.5 (air) - 25 (LSWT) -2.5 (maximum difference), which would give an cooling heat flux of about -3 W m-2 (using the formula in section 3.1). Therefore, even on the mean, the radiation effect might still be relevant and comparable in this case with the direct heat flux. Furthermore, it is not shown the partition between SW and LW. While LW radiation affects only the surface water temperature, SW penetrates the water column. I believe it is important to further detail the potential radiation effects. Figure 10 could be extended with two extra panels including SWnet and LWnet complementing the information in figure 9 to clarify potential impact of radiation, in particular solar in the differences between DWET and VWET.

The authors suggest that rainfall temperature and rain-induced turbulence could be implemented into lake models as a way to represent the effects of rainfall in LSWT. However, they do not show if a lake model (or several) are not able to represent the LSWT differences seen in the observations. Considering the high quality and length of the observations, simulations with a lake model in stand-alone model would prove

fundamental to support the authors suggestions. For example: does the model when forced with the observations also gives lake surface temperature differences comparable with the observations? This would strong support the efforts to represent missing processes. Another conclusion could be that other errors in the model have a higher impact and role of rainfall on LSWT is of secondary. I understand that this would require an extra and significant amount of work, and leave this decision to the editor in case the authors do not have the time and/or capacity to perform those simulations in a reasonable time window. If this is the case, I would encourage the authors to at least extend a bit more the conclusions suggesting model protocols to access this problem,

---

## Author Comment (AC1) · 14 Nov 2018

**Response to Referee 1**

We would like to thank the referee for their comments.

**SPECIFIC COMMENTS**

**The authors propose four possible mechanisms for the observed cooling. The first—near surface evaporative cooling of raindrops— is noted as a purely atmospheric phenomenon, and thus excluded from further discussion as the primary interest seems to be lake model parameterizations (Pg. 3, lines 18–19). This exclusion seems a bit hasty: there is no indication in the title emphasizing lake modelling, and the abstract explicitly states the importance for atmospheric**

**models (Pg. 1, lines 10–12). A little more discussion of this mechanism is warranted or the target audience should be identified as lake modelers in order to remove this slight inconsistency.**

Yes, we agree that some discussion of the evaporative cooling process should be added. In addition, we propose changing the end of the abstract to:

"...and suggests that further efforts are needed to quantify this effect in other regions and to include this process in **land-surface models used for atmospheric prediction**."

**The scaling analysis for mechanical mixing is interesting, but there is an underlying assumption that all of the incoming mechanical energy (i.e. the kinetic energy of the falling rain drops) is converted into TKE available for mixing. Is this process really 100% efficient? I expect some of the energy must go nearly instantaneously into heat through viscosity. A slightly more sophisticated scaling argument might be able to show that this fraction of energy is very small. At least the possibility should be mentioned.**

We can extend the discussion of the likely processes and mixing associated with raindrops, and include additional references. We can also revise the analysis to incorporate the effect of <100% efficiency in the conversion process.

**Radiometers have been used for measuring lake surface temperatures for many years, but there may be slightly different systematic errors under different conditions (i.e. heavy rain vs dull day). Since the temperature effect is relatively small, this might be important and worth a mention. There may be literature on this subject. Any comparison of the radiometric temperatures measured at this site with actual in situ instruments would be helpful — but I gather there are no in situ data? Note that the authors have addressed random measurement errors but not the possibility of different systematic errors.**

Yes we agree that the possibility of weather-related systematic errors should be added to the discussion. In this context we would remark that, while the rain rate during VWET

days is highest during the final few hours of the day, it may be seen from figure 5 that the average VWET rain rate is also appreciably higher than that of the DWET category at some earlier times. However the LWST differences, and their significance, at these earlier times are both less than during the final few hours of the day, as shown in figures 10 and 11.

Unfortunately we have not yet found any literature specifically on this subject which could be referenced. Nor were there any in-situ lake temperature measurements, as the referee has remarked. As indicated in the Discussion and Conclusion (p.8, lines 14–25), we would be greatly interested in obtaining such measurements in future if possible.

**Fig. 9 shows that during daytime the net radiation from WET exceeds VMET by quite a lot, presumably due to more SW radiation reaching the surface. This is the justification for introducing the DWET category. Fig. 9 also shows that during the night (16–5 UTC) the net radiation (i.e. net LW) for VWET exceeds both that for WET and DWET (by 7 or 8 W/m2)–implying either a colder surface or increased downward LW for VWET. This period largely coincides with the period of greatest rainrates for VWET shown in Fig. 5. It seems to me this strengthens the authors' argument that heavy rainfall cools the surface, but they don't point this out. This may be worth mentioning. Fig. 11 also shows the strongest signal is during 16–24 UTC.**

The referee makes an interesting point. We would relate this to the fact that, to err on the side of caution, the integrated radiation threshold for DWET days was chosen so that VWET days absorbed 1–2% more radiation in the mean than DWET days (see p.7, lines 12–13), yet they still show a colder LWST by the end of the day. We can emphasise this in a new version of the manuscript.

**MINOR COMMENTS AND FIGURES**

**Page 3, line 31: This claim needs a reference.**

We can expand this comment and add references. In doing so we would distinguish between rain effects on land which have been widely studied, and those in water bodies, which are less well observed or understood, and hence still debatable.

**Page 8, lines 16-18: Not clear why you have ruled out mechanism 1 (evaporative cooling)**
We would not say that we have ruled this mechanism out, but rather that the present study shows that the various other rain effects may also have an impact. We can alter the text to hopefully express this point more clearly.

**Fig. 1: would benefit from a context map inset showing location of the lake within Africa**
We agree and will be able to add a context map.

**Fig. 2: what are the large boxes in the upper left-hand image?**
These are metal containers housing a lab and technical equipment. We can add information on this in the figure caption, and also in the main text in section 4.1.

**Fig. 10: there seems to be a large jump between 23 UTC and 0 UTC. This does not seem physically possible, especially where the curves are changing smoothly. Is there a processing error?**
The reason for this is that the different categories of day are interspersed throughout the dataset. Thus, there are no jumps for the category ALL, but other days may be preceded and followed by days of different categories in different proportions. For example, it is apparent (but not yet stated in the text) that a higher proportion of DRY days precede VWET days than follow them, hence the temperature at the start of the mean VWET day is higher than at the end. We can add information on this aspect of the data, which may also be interesting in its own right.

---

## Author Comment (AC2) · 14 Nov 2018

**Response to Referee 2**

We would like to thank the referee for their comments.

**Section 3, ln 15: The authors point 4 processes in which rainfall can affect lake surface temperature. Since the authors mention the evaporative cooling, the solar radiation shading during daytime associated with clouds could be also mentioned as a process which should be, in principle properly represented by the atmospheric model.**

We agree that modelling the cloud cover correctly is extremely important for the purposes of predicting LWST. Since this study is primarily concerned with the effects of

rainfall, in the present context we would see cloud cover as a related atmospheric process. That is, significant rainfall requires the presence of clouds, but the presence of clouds does not guarantee rain. This is unlike the evaporative cooling referred to, which only occurs when rain is present.

We would therefore be happy to add this point about cloud cover, but would describe this as a related process rather than a direct influence of rainfall.

**Figure 4: Power spectrum of wind: There are several peaks on the sub-daily frequencies. Could the authors provide the frequencies of these and comment on their source (breeze effects?)**
The sub-daily wind fluctuations giving rise to these peaks are also evident on the wind-speed panels of figures 6 and 10. We assume these are due to local circulations caused by lake- or land-breezes, and the bi-modal distribution of the wind direction histograms (figure 7) would seem to also support this interpretation.

We can add information on the frequencies of the dominant sub-daily spectral peaks and their likely origin.

**The authors filtered the effect of radiation by defining the DWET days as days with net radiation below 1.5x10\*\*7 J m-2. The average different between DWET net radiation and VWET is about -2.3 W m-2. Visual inspection of T and LSWT mean diurnal cycles for VWET suggests a temperature difference between air and LSWT of about 22.5 (air) - 25 (LSWT) -2.5 (maximum difference), which would give an cooling heat flux of about -3 W m-2 (using the formula in section 3.1). Therefore, even on the mean, the radiation effect might still be relevant and comparable in this case with the direct heat flux.**
There may be some slight confusion here, so it may be worth beginning our response by repeating the relevant text from p.7:
*"With a threshold total radiation of $1.5 \times 10^7$ J m$^{-2}$, the average total radiation of DWET days is approximately $1.03 \times 10^7$ J m$^{-2}$, compared to an average total radiation of*

*1.05×10⁷ J m⁻² on VWET days."*

That is, the threshold has been chosen with a deliberate slight element of caution, so that VWET days absorb 1–2% *more* net radiation on average than DWET days. In our judgement this makes the later LWST cooling of VWET days relative to DWET days *less* attributable to a simple difference in net radiation.

We can add a further sentence to the text quoted above to emphasise the point made in this response.

**Furthermore, it is not shown the partition between SW and LW. While LW radiation affects only the surface water temperature, SW penetrates the water column. I believe it is important to further detail the potential radiation effects. Figure 10 could be extended with two extra panels including SWnet and LWnet complementing the information in figure 9 to clarify potential impact of radiation, in particular solar in the differences between DWET and VWET.**
As suggested, we are happy to extend figure 10 with extra panels showing the net LW and SW.

**The authors suggest that rainfall temperature and rain-induced turbulence could be implemented into lake models as a way to represent the effects of rainfall in LSWT. However, they do not show if a lake model (or several) are not able to represent the LSWT differences seen in the observations. Considering the high quality and length of the observations, simulations with a lake model in stand-alone model would prove fundamental to support the authors suggestions. For example: does the model when forced with the observations also gives lake surface temperature differences comparable with the observations? This would strong support the efforts to represent missing processes. Another conclusion could be that other errors in the model have a higher impact and role of rainfall on LSWT is of secondary. I understand that this would require an extra and significant amount of work, and leave this decision to the editor in case the authors do not have the time and/or capacity to perform those simulations in a reason-**

**able time window. If this is the case, I would encourage the authors to at least extend a bit more the conclusions suggesting model protocols to access this problem,**
Unfortunately, as the referee has anticipated, we do not have the available resource to perform a modelling study to go alongside this observational analysis. We would however be willing to discuss this in more detail as potential future work in the Discussion and Conclusion, which is the referee's alternative suggestion.

---

## Author Response (AR1)

**Response to Referee 1**

We would like to thank the referee for their comments.

**SPECIFIC COMMENTS**

**1. The authors propose four possible mechanisms for the observed cooling. The first—near surface evaporative cooling of raindrops—is noted as a purely atmospheric phenomenon, and thus excluded from further discussion as the primary interest seems to be lake model parameterizations (Pg. 3, lines 18–19). This exclusion seems a bit hasty: there is no indication in the title emphasizing lake modelling, and the abstract explicitly states the importance for atmospheric models (Pg. 1, lines 10–12). A little more discussion of this mechanism is warranted or the target audience should be identified as lake modelers in order to remove this slight inconsistency.**

We have added discussion of the evaporative cooling process as the new Section 3.1 (Cooling of near-surface air). In addition, the end of the abstract has been changed to:

"...and suggests that further efforts are needed to quantify this effect in other regions and to include this process in **land-surface models used for atmospheric prediction**."

**2. The scaling analysis for mechanical mixing is interesting, but there is an underlying assumption that all of the incoming mechanical energy (i.e. the kinetic energy of the falling rain drops) is converted into TKE available for mixing. Is this process really 100% efficient? I expect some of the energy must go nearly instantaneously into heat through viscosity. A slightly more sophisticated scaling argument might be able to show that this fraction of energy is very small. At least the possibility should be mentioned.**

Section 3.3 (Mechanical and convective mixing) has been extended to discuss the likely processes and mixing associated with raindrops, along with additional references. We have also revised the analysis in Section 3.3 to incorporate the effect of $<100\%$ efficiency in the conversion process. This has been incorporated using the efficiency fraction $\phi$. It may be seen that the main scaling calculation remains the same to the accuracy shown, for efficiencies in the range $0.5 \leq \phi \leq 1$. This result has been added to the text

(§ 3.3, penultimate paragraph).

**3. Radiometers have been used for measuring lake surface temperatures for many years, but there may be slightly different systematic errors under different conditions (i.e. heavy rain vs dull day). Since the temperature effect is relatively small, this might be important and worth a mention. There may be literature on this subject. Any comparison of the radiometric temperatures measured at this site with actual in situ instruments would be helpful — but I gather there are no in situ data? Note that the authors have addressed random measurement errors but not the possibility of different systematic errors.**

The possibility of weather-related systematic errors has been added to the discussion as a new paragraph at the end of Section 4.4 (Further partitioning by rainfall and net radiation). We have also briefly discussed there what can be inferred about these errors in the present case, given the different rain rates at different times of day.

**4. Fig. 9 shows that during daytime the net radiation from WET exceeds VMET by quite a lot, presumably due to more SW radiation reaching the surface. This is the justification for introducing the DWET category. Fig. 9 also shows that during the night (16–5 UTC) the net radiation (i.e. net LW) for VWET exceeds both that for WET and DWET (by 7 or 8 W/m2)–implying either a colder surface or increased downward LW for VWET. This period largely coincides with the period of greatest rainrates for VWET shown in Fig. 5. It seems to me this strengthens the authors' argument that heavy rainfall cools the surface, but they don't point this out. This may be worth mentioning. Fig. 11 also shows the strongest signal is during 16–24 UTC.**

As indicated previously in our response to this point, we have added the following text at the end of the first paragraph of Section 4.4 (Further partitioning by rainfall and net radiation):

"It is worth emphasising that the radiation threshold for DWET days was deliberately chosen to allow a small margin, so that VWET days absorbed 1–2% more radiation in the mean than DWET days. Thus any extra LWST cooling in VWET days compared to DWET days would be against a background of a slight excess of absorbed radiative energy on VWET days, in the

mean."

**MINOR COMMENTS AND FIGURES**

**5. Page 3, line 31: This claim needs a reference.**
We have added the following text to the end of Section 3.2 (Direct heat flux):
"On land, this process has been widely studied (Seneviratne et al., 2010; Taylor et al., 2012; Guillod et al., 2015; Lorenz et al., 2016). The rain effect in water bodies is more uncertain, although recently there have been some interesting observational studies e.g. Reverdin et al. (2012)."

**6. Page 8, lines 16-18: Not clear why you have ruled out mechanism 1 (evaporative cooling)**
As indicated previously, we did not intend ruling our evaporative cooling, and so the sentence in question in Section 5 has been changed to clarify this, as follows:
"However, through our indirect analysis of the processes it seems likely that cooling by rain combined with mechanical and convective mixing from droplet impact may have an effect on LWST, in addition to the effect from the more widely studied pathway of evaporative cooling."

**7. Fig. 1: would benefit from a context map inset showing location of the lake within Africa**
This has been added.

**8. Fig. 2: what are the large boxes in the upper left-hand image?**
These are metal containers housing a lab and technical equipment. Information on this has been added to the figure caption, and also in the main text in section 4.1.

**9. Fig. 10: there seems to be a large jump between 23 UTC and 0 UTC. This does not seem physically possible, especially where the curves are changing smoothly. Is there a processing error?**
The reason for this is that the different categories of day are interspersed throughout the dataset. Thus, there are no jumps for the category ALL, but other days may be preceded and followed by days of different categories in different proportions. We have added information on this aspect of the data in a new table, Table 2. The new table is referenced in the text in Section 4.4 (Further partitioning by rainfall and net radiation).

**Response to Referee 2**

We would like to thank the referee for their comments.

**1. Section 3, ln 15: The authors point 4 processes in which rainfall can affect lake surface temperature. Since the authors mention the evaporative cooling, the solar radiation shading during daytime associated with clouds could be also mentioned as a process which should be, in principle properly represented by the atmospheric model.**

As indicated previously in our response to this point, we would see cloud cover as a related atmospheric process, in the context of the present rainfall-based study.

In the first paragraph of Section 3 (Physical effects of Rainfall) we have extended the relevant sentence referring to the first process mentioned, evaporative cooling, so that it now reads:
"The first of these ought to be parametrised by atmospheric models, as with related atmospheric effects like the reduction of insolation by cloud cover."

**2. Figure 4: Power spectrum of wind: There are several peaks on the sub-daily frequencies. Could the authors provide the frequencies of these and comment on their source (breeze effects?)**

We have added this information, and some more discussion on the origin of these peaks, as a new paragraph at the end of Section 4.2 (Weather and climate).

**3. The authors filtered the effect of radiation by defining the DWET days as days with net radiation below 1.5x10**7 J m-2. The average different between DWET net radiation and VWET is about -2.3 W m-2. Visual inspection of T and LSWT mean diurnal cycles for VWET suggests a temperature difference between air and LSWT of about 22.5 (air) - 25 (LSWT) -2.5 (maximum difference), which would give an cooling heat flux of about -3 W m-2 (using the formula in section 3.1). Therefore, even on the mean, the radiation effect might still be relevant and comparable in this case with the direct heat flux.**

As indicated previously, we believe there may be some slight confusion here. The radiation threshold has been chosen with a deliberate slight element of caution, so that VWET days absorb 1–2% *more* net radiation on average

than DWET days.

To emphasise that the chosen threshold makes it slightly more difficult for VWET days to display the extra cooling observed, we have added the following text at the end of the first paragraph of Section 4.4 (Further partitioning by rainfall and net radiation):
"It is worth emphasising that the radiation threshold for DWET days was deliberately chosen to allow a small margin, so that VWET days absorbed 1–2% more radiation in the mean than DWET days. Thus any extra LWST cooling in VWET days compared to DWET days would be against a background of a slight excess of absorbed radiative energy on VWET days, in the mean."

**4. Furthermore, it is not shown the partition between SW and LW. While LW radiation affects only the surface water temperature, SW penetrates the water column. I believe it is important to further detail the potential radiation effects. Figure 10 could be extended with two extra panels including SWnet and LWnet complementing the information in figure 9 to clarify potential impact of radiation, in particular solar in the differences between DWET and VWET.**
We have extended figure 10 with extra panels showing the net LW and SW.

**5. The authors suggest that rainfall temperature and rain-induced turbulence could be implemented into lake models as a way to represent the effects of rainfall in LSWT. However, they do not show if a lake model (or several) are not able to represent the LSWT differences seen in the observations. Considering the high quality and length of the observations, simulations with a lake model in stand-alone model would prove fundamental to support the authors suggestions. For example: does the model when forced with the observations also gives lake surface temperature differences comparable with the observations? This would strong support the efforts to represent missing processes. Another conclusion could be that other errors in the model have a higher impact and role of rainfall on LSWT is of secondary. I understand that this would require an extra and significant amount of work, and leave this decision to the editor in case the authors do not have the time and/or capacity to perform those simulations in a reasonable time window. If this is the case, I would encourage the authors to at least extend a**

**bit more the conclusions suggesting model protocols to access this problem,**
As indicated previously, unfortunately we do not have the available resource to perform a modelling study to go alongside this observational analysis.

To take up the referee's alternative suggestion, we have added new text, comprising the penultimate paragraph of Section 5. Here we discuss ways in which lake model output may possibly be examined or reprocessed, in order to highlight any correlations between model error and rainfall.

**Manuscript changes as shown in the marked-up manuscript**

Abstract:

Wording changed at the end (Ref.1 comment 1)

p3:

Lines 4-5 have new references added to recently published papers in the special addition.

Lines 19-21 are changed to mention another, related process (Ref.2 comment 1)

New Section 3.1 added (Ref.1 comment 1)

p4:

Lines 12-14 add more discussion and references (Ref.1 comment 5)

Lines 18-32 extend the discussion of rain processes (Ref.1 comment 2)

p5:

Lines 1-22 include changes to text and analysis (Ref.1 comment 2)

p6:

Lines 3-6 describe the metal containers and instrument mounting (Ref.1 comment 8)

p7:

Lines 4-8 discuss the sub-daily peaks in the wind power spectra (Ref.2 comment 2)

p8:

Lines 6-9 emphasise the conservative nature of the radiation threshold (Ref.1 comment 4, Ref.2 comment 3)

Lines 10-12 introduce new Table 2, which adds information on how the cateogries precede and follow one another (Ref.1 comment 9)

p9:

Lines 4-11 discuss the possibility of systematic error (Ref.1 comment 3)

p10:

Table 2 has been added (Ref.1 comment 9)

Line 3 has been extended to include another process (Ref.1 comment 6)

Lines 4 – p11 line 6 have been added to discuss in more detail how future or other work could examine rainfall processes and assess their effects (Ref.2 comment 5)

p11:

Lines 11-13 have had text moved earlier to facilitate the extended future-work discussion.

Line 14 adds a statement on data availability.

Figure 1 has a small map added to provide context at the African scale (Ref.1 comment 7)

Figure 2 has reference to the metal containeres added (Ref.1 comment 8)

Figure 10 has new panels (e) and (f) added to show the net LW and SW mean evolution (Ref.2 comment 4)

**Estimating the effect of rainfall on the surface temperature of a tropical lake**

Gabriel Gerard Rooney[1], Nicole van Lipzig[2], and Wim Thiery[3,4]

[1]Met Office, FitzRoy Road, Exeter, EX1 3PB, UK
[2]KU Leuven, Department of Earth and Environmental Sciences, Celestijnenlaan 200E, 3001 Leuven, Belgium.
[3]ETH Zurich, Institute for Atmospheric and Climate Science, Universitaetstrasse 16, 8092 Zurich, Switzerland
[4]Vrije Universiteit Brussel, Department of Hydrology and Hydraulic Engineering, Pleinlaan 2, 1050 Brussels, Belgium

**Correspondence:** G. G. Rooney (gabriel.rooney@metoffice.gov.uk)

**Abstract.** We make use of a unique high-quality, long-term observational dataset on a tropical lake to assess the effect of rainfall on lake surface temperature. The lake in question is Lake Kivu, one of the African Great Lakes, and was selected for its remarkably uniform climate and availability of multi-year, over-lake meteorological observations. Rain may have a cooling effect on the lake surface by lowering the near-surface air temperature, by the direct rain heat flux into the lake, by mixing the lake surface layer through the flux of kinetic energy, and by convective mixing of the lake surface layer. The potential importance of the rainfall effect is discussed in terms of both heat flux and kinetic-energy flux. To estimate the rainfall effect on the mean diurnal cycle of lake surface temperature, the data are binned into categories of daily rainfall amount. They are further filtered based on comparable values of daily mean net radiation, which reduces the influence of radiative-flux differences. Our results indicate that days with heavy rainfall may experience a reduction in lake surface temperature of approximately 0.3 K by the end of the day compared to days with light-to-moderate rainfall. Overall this study highlights a new potential control on lake surface temperature, and suggests that further efforts are needed to quantify this effect in other regions and to include this process in  land-surface models used for atmospheric prediction.

*Copyright statement.* The works published in this journal are distributed under the Creative Commons Attribution 4.0 License. This licence does not affect the Crown copyright work, which is re-usable under the Open Government Licence (OGL). The Creative Commons Attribution 4.0 License and the OGL are interoperable and do not conflict with, reduce or limit each other.

© Crown copyright 2018

[revised manuscript text omitted]